# Association between the Prevalence of Indigestible Foreign Objects in the Gastrointestinal Tract of Slaughtered Cattle and Body Condition Score

**DOI:** 10.3390/ani7110080

**Published:** 2017-10-30

**Authors:** Vikhaya Vincent Nongcula, Leocadia Zhou, Kenneth Nhundu, Ishmael Festus Jaja

**Affiliations:** 1Department of Livestock and Pasture Science, University of Fort Hare, Alice 5700, South Africa; 200901532@ufh.ac.za; 2Risk and Vulnerability Science Centre, University of Fort Hare, Alice 5700, South Africa; lzhou@ufh.ac.za (L.Z.); knhundu@ufh.ac.za (K.N.); 3Agricultural Research Council, 1134 Park St, Hatfield, Pretoria 0001, South Africa

**Keywords:** environment, abattoir, cattle, indigestible foreign objects, waste, South Africa

## Abstract

**Simple Summary:**

The South African national cattle herd has increased from 6 million head since the 1970s to 14 million and more than two-thirds of the 14 million cattle in South Africa reside in the Eastern Cape Province (ECP), yet the Province has one of the highest poverty indexes. While this improvement is commendable, there are still many challenges to efficient livestock production notably infectious and non-infectious diseases. Many non-infectious diseases of the fore-stomach, such as rumenitis, rumen parakeratosis, traumatic reticuloperitonitis and poly bezoars, are rarely reported, but are known to obstruct the digestive functions of the fore-stomach, causing a marked reduction in animal weight, reproduction, feed absorption and productivity, and death. The pathogenesis of these diseases often begins with the consumption of indigestible foreign objects (IFOs). Animal husbandry in the ECP is mostly communal, with an extensive system of rearing animals as the most popular production system. This system of rearing animals exposes them to consumption of litter, especially when there is drought. This study provides knowledge on the prevalence, type and effect of the indigestible foreign object on the body condition score of animals slaughtered in the ECP.

**Abstract:**

It is estimated that South Africa’s population will be above 65 million in 2050. Thus, food production needs to triple to alleviate poverty and food insecurity. However, infectious and non-infectious diseases affect livestock productivity, thereby hampering food supply. Non-infectious disease/conditions caused by the consumption of solid waste material are rarely reported. Hence, this study investigates the occurrence and type of indigestible foreign objects (IFOs) in the stomach of slaughtered cattle in two high-throughput abattoirs (*n* = 4424) in the Eastern Cape Province of South Africa. The study revealed that metallic and non-metallic indigestible objects had an overall prevalence of 63% in cattle slaughtered in Queenstown abattoir (QTA, (*n* = 1906)) and 64.8% at the East London abattoir (ELA, (*n* = 2518)). Most of the IFOs were found in the rumen (64.2% and 70.8%) and reticulum (28.5% and 20.6%) at QTA and ELA respectively. The leading IFOs in the stomach of cattle at QTA were plastics (27.7%), poly bezoars (10.7%) and ropes (10.7%), while poly bezoars (19.8%), ropes (17.6%) and stones (10.7%) were the main IFOs seen in cattle at ELA. The study showed a statistical significance (*p* < 0.05) between body condition score and the prevalence of indigestible objects in cattle. The study concluded that litter and waste containing IFOs could pose a threat to livestock health and productivity. The practice of good animal husbandry and efficient solid waste management will mitigate the problem of animals consuming IFOs.

## 1. Introduction

A clean environment promotes good health, but the indiscriminate disposal of waste, including plastics, wood, papers, nails, rubber, and ropes, has a wide range of implications on the environment, and public and animal health. Livestock, especially ruminants, have been known to ingest indigestible matter (plastics, nails, rubber, wire, wood, ropes), especially during periods of drought [1,2]. An established reason why animals consume indigestible material is forage scarcity. Furthermore, cattle consume indigestible material such as plastic bags and metallic objects if their diet does not have enough essential minerals such as phosphorus. Hence, when under certain circumstances and exposed to litter, animals will consume such indigestible foreign objects. An adequate combination of vitamins and minerals in the correct ratio is important for growth and development in livestock [3,4]. None of the material previously mentioned can be digested by the ruminant stomach, and the consequences of these objects entering the digestive tract may include: sudden death, poly bezoars, traumatic reticulo-pericarditis, immuno-suppresion, bloat, indigestion, loss of condition and other unidentified health risks [5,6,7,8]. A poly bezoar is a stone-like material caused by a tightly packed collection of indigestible materials such as plastic, rope or hair combined and plant materials along with salt deposits that are trapped in the stomach or other parts of the gastrointestinal tract [9,10]. Livestock with poor body condition and inflammatory conditions like traumatic reticulo-pericarditis, immuno-suppression, and indigestion result in the carcass and offal downgrading or outright condemnation [11,12,13,14,15,16].

The impact of indigestible foreign objects (IFOs) on livestock health is devastating to the extent that surgical procedures and early-detection equipment are being widely adopted by farmers, nowadays [16]. However, such equipment is expensive, and out of reach for farmers in a developing country like South Africa. The Eastern Cape Province (ECP) has the highest number of livestock in the country, reared mostly by poorly resourced farmers under extensive farming [17] . For instance, out of the 14 million cattle in South Africa, over two-thirds reside in the ECP [18]. Many communal farmers keep livestock for multiple purposes, such as meat, milk, draught power, sale, as a source of income, and other functions [19,20]. Unlike livestock under commercial farming, animals in communal farming systems are not supplemented in times of droughts and feed scarcity. This consequently forces animals to search for food in unfamiliar sites, consuming any available feed including improperly disposed litter. Soji et al. [21] revealed that livestock, especially cattle, often roam and graze in profoundly polluted streets. Livestock deaths related to pollution have been reported as a limitation to production [22]. 

According to Dickson [23], the ECP lacks adequate and proper waste management system. The waste generated is easily accessible to animals, leading to a high number of mortalities, and non-digestible foreign bodies in cattle stomach. Also, animals are commonly seen grazing along major roads where there are many plastics and other waste products. Hence, the objective of the study was to determine the prevalence, type and effect of IFOs in cattle slaughtered in South Africa. 

## 2. Materials and Methods

### 2.1. Ethical Consideration

The study design and methods were approved by the University of Fort Hare ethics committee with reference number JAJ011SNON01. The research was carried out within the limits of the accepted ethical standards.

### 2.2. Study Site Description

The ECP (46.6%) is second to Limpopo Province (25.5%) in South Africa in having the lowest access to municipal waste services such as waste collection, treatment, recycling and disposal [24]. Municipal solid waste contributes a large percentage of the total waste generated in both urban and rural areas. Of the nine provinces, the ECP is ranked the poorest with regard to exposure to deprivation [25], with an estimated 61.6% and 52% households practicing poultry and livestock production, respectively [26]. The general climatic condition is shown in Figure 1 [27]. 

The study was carried out at two high-throughput abattoirs (QTA and ELA) situated in Queenstown and East London. Queenstown is under the Enoch Mgijima Local Municipality and is about 71.3 km^2^ with an altitude of 1081 m above sea level, located at 31°54′ S 26°53′ E. It lies on the Komani River close to the Wild Coast and Karoo Heartland. The climate is characterized by often extreme temperatures with figures sometimes reaching 35 °C in mid-summer and as low as −2 °C on a cold winter day. Queenstown normally receives about 399 mm of rain per year, with most rainfall occurring during summer. Rural households rely on natural resources [28] for their livelihoods. Livestock production is the predominant farming activity, and Queenstown is the centre for cattle trade in the Chris Hani District Municipality. Although the contribution of commercial agriculture to the economy has declined in recent years, livestock farming, particularly goats and cattle, has been listed as one of the priorities for investment in Queenstown [29]. 

East London is under the Buffalo city municipality, and is about 156.7 km^2^, located 32°59′ S 27°52′ E. In the Buffalo City municipality, agricultural land is chiefly owned by the white minority engaged in commercial farming, and has played a vital role in the surrounding areas, creating employment opportunities, generating income and alleviating poverty [30]. The climate is humid subtropical with low temperatures of 3 °C and the highest being 42 °C; the annual rainfall is about 593 mm per year, which mostly occurs in summer. ELA receives domesticated livestock species such as cattle, sheep, and pigs from throughout the province. 

### 2.3. Sampling Procedure

All animal handling (where applicable) was facilitated by the abattoir’s veterinarian and trained eviscerators. The abattoirs were visited between July and December 2015 during slaughter days to determine the occurrence and types of indigestible foreign objects, as well as to record the body condition score in slaughter cattle in QTA and ELA. Cattle were selected for this study because they are poor selective grazers and are more likely to ingest litter compared to sheep or goats [31]. Animal included in the study were selected using a systematic random sampling; details regarding the sample size and sampling methods were described by Jaja et al. [32]. Briefly, all animals were selected at equal intervals, except that the first animal was selected at random. This sampling method was used because of the large number of animals slaughtered per day and the constraints associated with abattoir work conditions.

### 2.4. Data Collection

The survey involves the documentation of the type and location of indigestible foreign objects (IFO) found in the gastrointestinal tract (GIT) of cattle after evisceration. Data on animal body condition scoring (BSC) was obtained antemortem using a modified scoring system of 1 (emaciated) to 5 (obese) according to the method described the by the University of Georgia, Cooperative Extension services [33]. Age estimation was done using dentition [34], and sex was recorded using visual appraisal. After slaughter, the contents of the forestomach and intestines were removed and carefully examined by trained eviscerators for the presence of IFOs. 

An inventory of the types of materials and their location in the cattle stomach was determined and subsequently analysed to determine the relationship between the BSC of the cattle and the occurrence of IFOs. 

### 2.5. Data Analysis

The types of IFOs obtained in the cattle stomach were documented alongside the occurrence and location. All data were managed using computer-based data management systems, Microsoft Excel 2010 and Statistical Package for Social Science (SPSS) version 23 (IBM Corporation, New York, NY, USA). The data were analyzed using frequency tables, percentages and charts as main descriptive indicators to show the relationship between sex, age, location and body condition score upon the occurrence of IFOs. Prevalence of IFOs was expressed as the number of cases of foreign bodies present to the total number of animals examined at the slaughter houses. The effect of IFOs on body condition score (BSC) was determined using Chi-square test; where BSC was the dependent variable and IFOs were the independent variables.

## 3. Results 

### Prevalence, Type, Location and Association of IFOs with Body Condition Scores of Slaughter Cattle in Various Ages at QTA and ELA

The main IFOs found in the forestomach of cattle were plastics, wire, nails, rope, bones, stones, clothes, hairs and poly bezoars. The study observed that plastics (27.7% and 10.7%), poly bezoars (10.7% and 19.8%) and ropes (7.6% and 17.6%) were the most abundant IFOs in slaughter cattle at abattoir QTA and ELA (Table 1). Other IFOs, such as nails, bones, clothes, etc., ranged between 2.6% and 5.0%. 

Table 2 shows that IFOs were found mostly in the rumen (64.2% and 28.5%) and reticulum (70.8% and 20.6%) for QTA and ELA, respectively. The least of the IFOs occurred in the abomasum (QTA = 7.3% and ELA = 5.9%) in both abattoirs. However, in ELA, it was observed that there were IFOs in the intestines (2.7%). At QTA, the highest incidence of IFOs was in cattle aged 3–5 years (47.0%), while at the ELA an incremental presence of IFO was observed in the various age brackets but peaked at five years of age (42.3%). More IFOs were found in females (66.2% and 56.1%) than in males (33.8% and 43.9%) in both QTA and ELA. 

From our findings, we observed the presence of IFOs in both animals with poor as well as those with good body condition. Cattle in average body condition (44.4% in QTA and 58.5% in ELA) were more likely to house IFOs than obese cattle (0.4%). There was an association (*p* < 0.05) between the prevalence of IFOs and BCS of cattle in both QTA and ELA. Table 2 depicts the relationship between BCS and the occurrence of IFOs. 

## 4. Discussions

To the best of our knowledge, there is no published research on the prevalence of indigestible foreign objects in cattle in South Africa. The current study revealed a high prevalence of IFOs in cattle slaughtered at two high-throughput abattoirs in the Eastern Cape Province of South Africa. The occurrence of indigestible objects has previously been reported as a threat to ruminant health [11,35,36] in Ghana, Nigeria and Ethiopia. 

We observed that non-metallic objects such as plastics and poly bezoars were the most numerous objects found in cattle at QTA, whereas poly bezoars and ropes were the most prevalent in cattle at ELA. The high plastic proportion found in QTA could be attributed to the fact that cattle from communal and semi-urban areas graze along major roads littered with plastic material. Furthermore, the Buffalo municipal area is proximal to the Atlantic Ocean, and frequently experiences strong winds that transport IFOs from one location to another. Animals in both commercial and communal farm land consume these materials during grazing. According to Getchell [37], livestock in resource-poor communities in South Africa frequently ingest plastic material in a bid to access food material enclosed in polyethene bags. In other parts of Africa, there have been similar reports of high prevalence of plastic in the stomach of cattle by Mohammed and Fromsa [6], who found 59.6% prevalence in Ethiopia, and Mushonga et al. [9], who reported 65.0% prevalence in Rwanda. Compared to the results obtained in this study, Negash et al. [38] found a much lower prevalence of poly bezoars (1%) in cattle of the Haramaya municipal abattoirs in Ethiopia. The occurrence of poly bezoars in high percentage in this study indicates early exposure of cattle to the indigestible material, as well as ingestion of more than one type of indigestible material. A poly bezoar is a stone-like material caused by indigestible materials such as plastic, rope or hair combined. It may also include plant materials along with salt deposits. 

In the ELA, the presence of ropes and plastics in study animals suggests that animals raised in an extensive system of farming tend to feed on indigestible material during periods of drought and feed scarcity [39]. Furthermore, animals reared under an intensive system of husbandry could experience sizeable amounts of plastics and other materials brought into the farm due seasonal dynamics such as strong winds in autumn and winter, which could expose them to these IFOs, especially if the animals have pica. Other objects, such as nail, bone, cloths and stones, found in this study are not uncommon among previously reported indigestible foreign materials in ruminants [36,39,40] in Ethiopia, Nigeria, and Kenya. The aforementioned studies considered these indigestible materials to be a threat to ruminant health and productivity. The occurrence of bones, as reported in this study, seems to be very scant in the literature. However, broken teeth, along with small pieces of bone were reported at 1.9% in the reticulum of Buffalo by Khan et al. [41] in Pakistan, but their significance was not discussed. 

Metallic indigestible objects found in QTA and ELA were nails and wires. Similar findings were reported by Anwar et al. [42], who found a 5.3% prevalence for both nails and wires in slaughter cattle in Pakistan. According to Ravindra et al. [43], cattle tend to swallow objects quickly during grazing and they do not use their lips for sensory purposes, hence the accidental ingestion of both metallic and non-metallic materials. Moreover, Tesfaye and Chanie [44], highlighted in their study that differences in solid waste management practices in various location where animals are reared, lack of proper animal husbandry such as dumping of metallic objects from old farm fences and construction sites, as well as materials used for packaging left on the farm, could lead to animals consuming IFO. Similarly, Pritchett and Whary [45], reviewing the epizootiology of hardware disease revealed that multiple cases of the disease might occur if feedlot or dairy cattle consume chopped feeds such as silage presented with shards of metal that would have been processed along with the feed. 

The rumen and reticulum were the main stomach compartments with a high prevalence of IFOs. The current finding concurs to those of Sheferaw et al. [2], who reported the prevalence of indigestible foreign materials in free-grazing ruminants in Ethiopia. The prevalence of indigestible foreign materials in rumen and reticulum could be attributed to the fact that ingested feed, along with indigestible material, enters directly into the rumen, and some indigestible material progresses to the reticulum. Furthermore, objects such as nails and stones are heavier than most swallowed objects. Thus they gravitate to the reticulum. A study by Herren, and Mushonga et al. [9,46] found no indigestible objects in the omasum of cattle slaughtered at two abattoirs. However, small stones were occasionally found in the intestines and abomasum in both abattoirs. Radostitis et al. [47] reported that abomasal impaction in beef suckler cows, dairy, and feedlot cattle could be due to the ingestion of gravel (combination of sand and small stones) possibly induced by either pica or the inadvertent mixture of gravel with the feed when it is scraped from bunker soils. A similar scenario could be possible in overgrazed pasture; grass tends to be shorter, and thus cattle can be exposed to the ingestion of small stones located at the grass base. Depending on size, stones can be transported along with ingested feed to the abomasum which may progress to the intestines. 

The present study revealed that older animals and female cattle were more susceptible to the ingestion of IFOs than young and male cattle. A similar result has been observed in other studies [9,48]. The common reasons for the high prevalence of IFOs in females and older animals include nutritional stress during gestation and duration of exposure to areas with the indigestible foreign material. Female animals are seldom considered for slaughter except in cases of mastitis, poor reproductive performance, poor body condition and age [49]. 

In this study, most of the IFOs were found in thin and average cattle. The difference in the body condition scores between cattle slaughtered at the two abattoirs could be attributed to the fact that the East London abattoir receives more cattle for slaughter from intensive or feedlot farming systems than the QTA. Livestock under intensive farming are supplemented with feed especially during periods of drought whereas, cattle under the extensive farming system are not supplemented with feed [4]. In addition, the proximity of the study area to the coastal line and presence of strong wind that disperses improperly disposed garbage to vulnerable animals further promotes the consumption of IFOs. The aforementioned reason could have been responsible for the differences in the occurrence of IFOs in this study. Regardless of this factor, Mushonga et al. [9] highlighted the possibility of indigestible plastics interfering with food digestion and absorption, leading to poor body condition. Therefore, it is difficult to make conclusions without further investigation.

The types of IFOs encountered in this study were plastics, wire, nails, rope, cloth, bones, hair, stone and poly bezoars. Statistical analysis indicated that the IFOs had an association with body condition scoring of affected cattle. Most of the IFOs in the current study were plastics, and the results suggest an association between thin and average body scoring condition of cattle and IFOs in both abattoirs. Even though the present study did not consider the medical history of the study animals to evaluate the implication IFO, Mushonga et al. [9] argued that the poor condition of animals may be a result of plastics interfering in food digestion and absorption. Tiruneh and Yesuwork [48], also attributed reduced weight gains in emaciated and thin animals to the interference of indigestible foreign bodies with volatile fatty acids. The presence of ropes in cattle slaughtered at ELA may have directly or indirectly resulted in weight loss reported in thin cattle, unlike in QTA, where the results suggest otherwise. The presence of ropes in animals that were slaughtered at ELA is suggestive of improper disposal of ropes used for cattle restraint or ropes used by the surrounding residents. Other IFOs, such as hairs, stones, and clothes, reported in this study were also statistically significant in body condition scoring of affected cattle, with results indicative of thin to average BSC. To further buttress the dangers associated with the consumption of IFOs, Eze [50], reported that ruminal obstruction might be due to impaction of nylon, ropes, and clothes in a five-year-old cow resulting in complications such as difficult respiration, anorexia, and recumbency. This study reported the presence of similar objects and suggests the possibility of study animals experiencing some of the aforementioned clinical signs. Although the association between hunger and thin BCS was not evaluated in this study, it is possible that the recovered IFOs could interfere with the normal function of the digestive system. In addition, inflammatory response to poly bezoar formation and blockage of food passage could lead to weight loss, rumenotomy and death [51].

Metallic IFOs reported in the present study were also associated with poor body conditions in QTA, but not in ELA. This difference can be attributed to the fact that metals are usually recycled, and recycling programs are scanty in rural areas. Regardless of this difference, the presence of metals in the fore-stomach could lead to inflammatory conditions commonly associated with traumatic reticulopathy. This condition are often resolved with surgical procedures such as remenotomy [16]. Therefore, our study suggests that metallic and non-metallic indigestible materials could lead to poor body condition of the study cattle, which is a consequence of its interference with digestion of food. The presence of metallic and non-metallic objects in cattle stomach may have direct and indirect effects on livestock productivity which could further hinder the sustainable production of food to mitigate hunger. 

## 5. Conclusions

The study revealed the prevalence of indigestible foreign objects in cattle slaughtered in two high-throughput abattoirs of the Eastern Cape Province. The rumen and reticulum harboured most of the IFOs. Animals that harboured IFOs in this study were associated with a poor body condition score. Although this study did not investigate the cellular, metabolic, anatomical and obstructive changes caused by the presence of IFO in the study animals, the overarching evidence in the literature and categorization of IFOs by BCS in the present study, denotes a strong direct or indirect relationship between the presence of IFOs and animal poor body condition score. The impact of animal weight loss and consequent poor body condition on the overall performance of livestock and food security occasioned by environmental contaminants such as IFOs is dire. There is no one-size-fits-all solution. However, good animal husbandry, including proper fencing of farms, supplementation of feed, and general biosecurity steps will enhance livestock productivity. A better system of solid waste management is crucial to mitigating livestock access to indigestible materials. Government incentives to farmers in the form of feed, farm fence and soft loans will promote good agricultural practices. Apart from providing all the incentives previously mentioned, government should continue and complete the on-going fencing of farm lands bordering major roads in the province. The fencing will limit animals to grazing on communal grazing land, rather than roads where they get the chance to consume IFOs. This study is the first of its kind in the country, and can serve as a basis for further research on the direct and indirect association between indigestible foreign bodies and animal loss of body condition. This study suggests that further research should be conducted on animals’ physiological and cellular response to the presence IFOs and, their attendant implications for weight gain.

## Figures and Tables

**Figure 1 animals-07-00080-f001:**
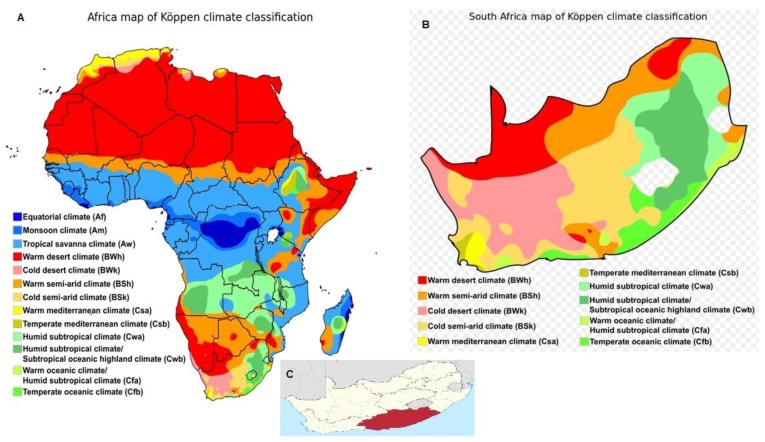
(**A**) Map of Africa of Koppen climate classification system; (**B**) Map of South Africa; (**C**) Map of the Eastern Cape Province (ECP) [27].

**Table 1 animals-07-00080-t001:** Type and prevalence of indigestible foreign objects at QTA (*n* = 1906) and ELA (*n* = 2518).

IFO	QTA			ELA		
	NCHIFO	Prevalence (%)	# BCS	NCHIFO	Prevalence (%)	# BCS
Plastic	528	27.7	0.00 *	270	10.7	0.00 *
Wire	66	3.5	0.00 *	0	0.0	ND
Nail	30	1.6	0.05 *	66	2.6	0.40
Rope	144	7.6	0.15	444	17.6	0.00 *
Hair	96	5.0	0.35	0	0.0	ND
Cloth	66	3.5	0.01 *	84	3.3	0.68
Stone	0	0.0	0.22	270	10.7	ND
Bone	66	3.5	0.00 *	0	0.0	0.00 *
Polybezoars	204	10.7	0.00 *	498	19.8	0.17
Total	1200			1632		

Note: QTA; Queenstown abattoir, ELA; East London abattoir, IFO; Indigestible foreign objects, # BCS; Chi-square significance calculated according to body condition score, * Significant at *p* < 0.05; ND (No Data), NCHIFO means the number of cattle harbouring different type of IFOs. In this instance, some cattle harboured more than one type of IFO. Hence the total number of documented IFO exceeded the number of animals sampled.

**Table 2 animals-07-00080-t002:** Number of IFOs according to location, age, sex and body condition scoring in cattle slaughtered in QTA and ELA.

Category		QTA (*n* = 906)		ELA (*n* = 1518)	
		NCHIFO	Prevalence (%)	NCHIFO	Prevalence (%)
Stomach	Rumen	582	64.2	1074	70.8
	Reticulum	258	28.5	312	20.6
	Omasum	0	0	0	0
	Abomasum	66	7.3	90	5.9
	Intestines	0	0	42	2.7
Total		906	100	1518	100
Age	1–3	186	20.5	300	19.8
	3–5	426	47.0	576	37.9
	>5	294	32.5	642	42.3
Total		906	100	1518	100
Sex	Male	306	33.8	666	43.9
	Female	600	66.2	852	56.1
Total		906	100	1518	100
BCS					
1	Emaciated	0	0	0	0
2	Thin	300	33.1	246	16.2
3	Average	402	44.4	888	58.5
4	Fat	204	22.5	378	24.9
5	Obese	0	0	6	0.4
Total		906	100	1518	100

NCHIFO means the number of cattle harbouring different type of IFOs.

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
