# Peer review of "Association between the Prevalence of Indigestible Foreign Objects in the Gastrointestinal Tract of Slaughtered Cattle and Body Condition Score"

_animals, 2017, doi:10.3390/ani7110080_

Round 1

Reviewer 1 Report

Manuscript: Nongcula et al., Association between the prevalence of indigestible 2 foreign objects in the stomach of slaughtered cattle 3 and body condition score.

This observational study that focuses on the incidence and content of IFOs present within cattle processed at two abattoirs in South Africa. This investigation provides some interesting observations that would benefit the veterinary and livestock industry not only within Africa, but globally.

During my review of the manuscript, I believe there are some issues within the paper that needs to be addressed and clarified.

Major concerns

2.4 Data collection: As body condition scoring is an important consideration within the manuscript, a more detailed description, perhaps within a table, of the body scoring system be provided. As well, please provide more detailed information how the scoring system was standardized between the two abattoirs.

Line149: Table 3 is missing. Data appears to be presented within Table 2. Please make the required adjustments.

 Figure 1: The image on the far right has a ‘checkered background’ and this is distracting. I would recommend editing  the background and replacing it with a ‘white background’.

Line 137: Data within the text for ploy bezoars has two values. It’s unclear within the text the source (ie .ELA or QTA) of the data. For clarity, all data within Results section 3.1 should identify the source (ie ETA or QTA).

Minor concerns:

Line 66: please define drought power.

Line 174: please define poly bezoars within the introduction, not the discussion section.

Author Response

Comments and Suggestions for Authors

Manuscript: Nongcula et al., Association between the prevalence of indigestible 2 foreign objects in the stomach of slaughtered cattle 3 and body condition score.

This observational study that focuses on the incidence and content of IFOs present within cattle processed at two abattoirs in South Africa. This investigation provides some interesting observations that would benefit the veterinary and livestock industry not only within Africa, but globally.

During my review of the manuscript, I believe there are some issues within the paper that needs to be addressed and clarified.

Major concerns

2.4 Data collection: As body condition scoring is an important consideration within the manuscript, a more detailed description, perhaps within a table, of the body scoring system be provided. As well, please provide more detailed information how the scoring system was standardized between the two abattoirs.

Response:

The authors are grateful to the reviewer for this important observation. The authors have provided details on how the body condition score was determined and the text currently read thus: Data on animal body condition scoring (BSC) was obtained ante mortem using a modified scoring system of 1 (emaciated) to 5 (obese) according to the method described the by the University of Georgia, Cooperative Extension services (NRC, 2000), line 135-139. The standardization was by means of the document already cited. We made sure that the criteria for animal selection into the various BCS groups were the same in both abattoirs.

Line149: Table 3 is missing. Data appears to be presented within Table 2. Please make the required adjustments.

Response:

The authors are grateful to the reviewer for this keen observation. Table 3 has been changed to table 2. See line 172.

 Figure 1: The image on the far right has a ‘checkered background’ and this is distracting. I would recommend editing the background and replacing it with a ‘white background’.

Response:

The authors are grateful to the reviewer for this keen observation. The authors have partly adjusted the background to make it plain. The challenge with figure 1 is that it is a diagram extrapolated with approval of the original owners (Jaja et al., 2017). We kindly request that the reviewer accept the improved figure.

Line 137: Data within the text for ploy bezoars has two values. It’s unclear within the text the source (ie .ELA or QTA) of the data. For clarity, all data within Results section 3.1 should identify the source (ie ETA or QTA).

Response:

The authors are grateful to the reviewer for this profound observation. We have amended the line 136 to 138 for clarity and it reads thus “The study observed that plastics (27.7% and 10.7%), poly bezoars (10.7% and 19.8%) and ropes (7.6% and 17.6%) were the most abundant IFO’s in slaughter cattle at abattoir QTA and ELA (Table 1).” In some animals more than one type of IFO occurred hence the total number of IFO’s exceeed the number of animals sampled.

Minor concerns:

Line 66: please define drought power.

Response:

The authors are grateful to the reviewer for this observation. We have changed the misspelled word from “drought power” to “draught power”.

Line 174: please define poly bezoars within the introduction, not the discussion section.

Response:

The authors are grateful to the reviewer for this observation. We have defined poly bezoars in line 59-62 as “A poly bezoar is a stone-like material caused by a tightly packed collection of indigestible materials such as plastic, rope or hair combined and plant materials along with salt deposits that are trapped in the stomach or other parts of the gastrointestinal tract”.

Reviewer 2 Report

This manuscript presents research that is of interest but in my opinion requires a major review before it is suited for publication.

Overall the clarity and grammatical correctness of the English language of the entire manuscript requires reviewing. For example the title should not just refer to 'stomach' but define the anatomical correct location of the ruminant gastrointestinal system.

The methods describes in great detail the study sites of the two abattoirs where the research was detected yet the relevance of this becomes not clear. However the sampling procedure is only minimally described and should clearly describe how many cattle were examined and when and how they were selected. At present this is not done. The result section refers only to cattle that had foreign bodies detected-I presume there were also some examined that had no foreign bodies detected. The total number of cattle examined needs to be stated including how many were examined that had no foreign bodies detected.

The method section should further clearly describe what data was collected and how it was collected or categorised.

The results section is not clear either. For example table 1 refers to n=906 for the QTA abattoir yet the numbers add up to 1200. The column headings are not clear. i.e. what is occurrence? Do you mean numbers? #BCS is actually p-value so should be labelled as such. The same applies for table2. Please define what you mean by occurrence i.e. is it number of animals detected?  The varied categories need units i.e. age is it years? BCS should be described by scores as you describe these in methods.

Ultimately at present the manuscript does not describe if the study found that the presence of foreign bodies is associated with lower body condition scores or not but only that certain types of foreign bodies appear to be associated with BCS.

Author Response

Comments and Suggestions for Authors

This manuscript presents research that is of interest but in my opinion requires a major review before it is suited for publication.

Overall the clarity and grammatical correctness of the English language of the entire manuscript requires reviewing. For example the title should not just refer to 'stomach' but define the anatomical correct location of the ruminant gastrointestinal system.

 Response

The authors are grateful to the reviewer for this observation. But we prefer that the word “stomach” be changed to “gastrointestinal tract” in order for the title to cater for the presence of indigestible foreign objects found in the intestines.  The intestine strictly is not part of the stomach but the GIT. We kindly request the reviewer to accept our position. Authors sent the manuscript to a professional English editor at the teaching and learning center of the University for editing. Her views and concerns have been incorporated to the current manuscript.

The methods describes in great detail the study sites of the two abattoirs where the research was detected yet the relevance of this becomes not clear. However the sampling procedure is only minimally described and should clearly describe how many cattle were examined and when and how they were selected. At present this is not done. The result section refers only to cattle that had foreign bodies detected-I presume there were also some examined that had no foreign bodies detected. The total number of cattle examined needs to be stated including how many were examined that had no foreign bodies detected.

Response

The authors grateful to the reviewer for this observation. The authors had hoped to highlight the study area to reflect the climate and seasons in order for readers to understand how the climate of the region can affect the availability of IFO’s. The sampling procedures had further been expanded to describe the sampling method and method of data collection, see line 127-136 and 157-159. Also see table 1 and 2. Animal included in the study were selected using a systematic random sampling, details regarding the sample size and sampling methods were described by Jaja et al. [34]. Briefly, all animal were selected at equal intervals except that the first animal was selected at random

The method section should further clearly describe what data was collected and how it was collected or categorised.

Response

The authors thank the reviewer sincerely for this crucial observation. In stating what data was collected and how the data was categorised, the authors have included in line 135-139 “the survey involves the documentation of the type and location of foreign indigestible bodies (IFO) found in the gastrointestinal tract (GIT) of cattle after evisceration. Also in line 129-132 “Animal included in the study were selected using a systematic random sampling, details regarding the sample size and sampling methods were described by Jaja et al. [34]. Briefly, all animal were selected at equal intervals except that the first animal was selected at random. This sampling method was used because of the large number of animals slaughtered per day and the constraint associated with abattoir work conditions”. Animals were categorized by age, sex and BCS.

The results section is not clear either. For example table 1 refers to n=906 for the QTA abattoir yet the numbers add up to 1200. The column headings are not clear. i.e. what is occurrence? Do you mean numbers? #BCS is actually p-value so should be labelled as such. The same applies for table2. Please define what you mean by occurrence i.e. is it number of animals detected?  The varied categories need units i.e. age is it years? BCS should be described by scores as you describe these in methods.

Response

The authors thank the reviewer sincerely for this crucial observation. Several aspect of the result has been re-written for clarity. We agree that the word occurrence as presented in table 1 and 2 is confusing, hence we decided to change it to “Number of  cattle habouring different IFO’s (NCHIFO)”. To further simplify the result we included an explanatory footnote so that reader will not be confused by the total number/type of IFO recovered from the GIT e.g for QTA 1200 and ELA 1632 which exceeded the number of animals sampled. The BCS score is included in table 2. How significant the BCS is to the documented IFO’s can be found in table 1. You will find this in line 157-163.

Ultimately at present the manuscript does not describe if the study found that the presence of foreign bodies is associated with lower body condition scores or not but only that certain types of foreign bodies appear to be associated with BCS.

Response

The authors thank the reviewer sincerely for this crucial observation. The authors included in the conclusion line 298-306 “Animals that harboured IFOs in this study were associated with poor body condition score. Although this study did not investigate the cellular, metabolic, anatomical and obstructive changes caused by the presence of IFO in the study animals, the overarching evidence in the literature and categorization of IFO’s by BCS in the present study, denotes a strong direct or indirect relationships between the presence of IFO’s and animal poor body condition score. The impact of animal weight loss and consequent poor body condition overall performance of livestock and food security occasioned by environmental contaminant such as IFO’s is dire.  

Round 2

Reviewer 1 Report

Authors rebuttal have adequately addressed this  reviewer' s comments

Author Response

Authors are grateful to the reviewer for a thorough review of our manuscript. We appreciate your comment and observations.

Reviewer 2 Report

line 11: insert 'the' before south African national cattle herd

line 14 delete 'but'

Section 2.4 line 131 : please state who did the examinations

line 195 suggest is spelt with a small s not capitol.

line 249 onwards in discussion section; this study design does not allow to suggest causality between low BCS and foreign bodies but only an association. For example is is quite likely that hungry cattle that are in low body condition score due to lack of nutrition are more likely to ingest foreign materials i.e. the poor nutrition caused the low BCS. This should be discussed in you discussion section in more detail. and I suggest to talk about an association of low BCS and ingestion of foreign material rather than causation.

Author Response

Reviewer 2:

Comments and Suggestions for Authors

line 11: insert 'the' before south African national cattle herd

Authors thank the reviewer for this observation. We have included “The” see line 11.

line 14 delete 'but'

Authors thank the reviewer for this observation. We have delete “but” see line 14.

Section 2.4 line 131 : please state who did the examinations

Authors thank the reviewer for this suggestion. We have stated in line 130-131 that “After slaughter, the contents of the forestomach and intestines were removed and carefully examined by trained eviscerators for the presence of IFOs”.

line 195 suggest is spelt with a small s not capitol.

Authors thank the reviewer for this observation. We have changed“Suggest to suggest” see line 195.

line 249 onwards in discussion section; this study design does not allow to suggest causality between low BCS and foreign bodies but only an association. For example is is quite likely that hungry cattle that are in low body condition score due to lack of nutrition are more likely to ingest foreign materials i.e. the poor nutrition caused the low BCS. This should be discussed in you discussion section in more detail. and I suggest to talk about an association of low BCS and ingestion of foreign material rather than causation.

Authors thank the reviewer for this observation and suggestion. Authors have adjusted the statement in line 252-253 to read “results suggest an association between thin and average body scoring condition of cattle and IFO’s in both abattoirs. In line 258-259 “The presence ropes in cattle slaughtered at ELA may have directly or indirectly resulted in thin cattle, unlike in QTA which the results suggest otherwise”. In line 267-271 “Although the association between hunger and thin BCS was not evaluated in this study, it is possible that the recovered IFO’s could interference with the normal function of the digestive system. In addition, inflammatory response to polybezoar formation and blockage of food passage could lead to weight loss, rumenotomy and death [54].

In the conclusion, the authors included “Apart from providing all the incentives previously mentioned, government should continue and complete the on-going fencing of farm lands bothering major roads in the province. The fencing will limit animals to grazing on communal grazing land rather than roads where they get the chance to consume IFO’s”. you will find this in line 305-308.